# European System for Cardiac Operative Risk Evaluation II and Liver Dysfunction

**DOI:** 10.3390/biomedicines13010154

**Published:** 2025-01-10

**Authors:** Andreea Ludusanu, Adelina Tanevski, Bogdan Mihnea Ciuntu, Razvan Lucian Bobeica, Dragos Andrei Chiran, Cristinel Ionel Stan, Viorel Dragos Radu, Vasile Lucian Boiculese, Grigore Tinica

**Affiliations:** 1Department of Morphofunctional Sciences I—Anatomy, University of Medicine and Pharmacy “Gr. T. Popa”, 700115 Iasi, Romania; andreealudusanu1106@yahoo.com (A.L.); dragos-andrei.chiran@umfiasi.ro (D.A.C.); cristinel.stan@umfiasi.ro (C.I.S.); 2Department of General Surgery, University of Medicine and Pharmacy “Gr. T. Popa”, 700115 Iasi, Romania; papancea.adelina@umfiasi.ro (A.T.); bogdan-mihnea.ciuntu@umfiasi.ro (B.M.C.); 3Department of Urology, University of Medicine and Pharmacy “Gr. T. Popa”, 700115 Iasi, Romania; dr.razvanbobeica@gmail.com; 4Biostatistics, Department of Preventive Medicine and Interdisciplinarity, University of Medicine and Pharmacy “Gr. T. Popa”, 700115 Iasi, Romania; lboiculese@gmail.com; 5Cardiac Surgery, University of Medicine and Pharmacy “Gr. T. Popa”, 700115 Iasi, Romania; grigoretinica@yahoo.com

**Keywords:** MELD score, EUROSCORE II, liver dysfunction, open-heart surgery, BACG, new risk score

## Abstract

**Background:** The importance of liver dysfunction in predicting mortality in patients undergoing cardiovascular surgery is an important topic due to the general desire to improve current risk scores such as EUROSCORE II (European System for Cardiac Operative Risk Evaluation), with EUROSCORE III being currently under development. The model for End-Stage Liver Disease (MELD) Score has already proven its utility in predicting outcomes for patients undergoing abdominal, cardiovascular or urological surgery. In the present study, we want to see its usefulness in proving the postoperative mortality in patients undergoing coronary artery bypass surgery. **Methods:** This was a retrospective study, and it included 185 patients, with 93 survivors being randomly chosen from a total of 589 surviving patients using age, emergency and the weight of cardiac procedures as criteria to match the 92 deceased patients during hospitalization in the postoperative period who underwent coronary artery bypass grafting (CABG) alone or CABG and other concomitant cardiovascular interventions during a 10-year period of time. We calculated for all these patients, at the time of admission, the MELD Score and EUROSCORE II, and we analyzed the predictive performance of the two scores and their constituents. **Results:** In the multivariable model, patients with a MELD Score ≥ 5.54 had a 2.38-fold increased risk of death (95% C.I.: 1.43–3.96, *p* = 0.001), while those with a EUROSCORE ≥ 10.37 had a 8.66-fold increased risk of death (95% C.I.: 3.09–24.29, *p* < 0.001). After combining the two scores, the conditional scenario achieved a high overall accuracy of 84.32% (*p* < 0.001) in predicting mortality. **Conclusions:** Patients with a MELD Score ≥ 5.54, had good sensitivity and a very good specificity in terms of mortality prediction, but the conditional scenario, leveraging both risk scores, i.e., the MELD Score and EUROSCORE, offers the highest utility in terms of enhancing mortality prediction regarding these patients.

## 1. Introduction

As the population changes and ages, the number of patients requiring cardiovascular surgery and showing signs of liver dysfunction has also increased in recent years [1]. Patients undergoing cardiac surgery who experience liver dysfunction face an elevated risk of both short-term and long-term mortality [2]. Additionally, these individuals are more likely to experience various adverse events during the perioperative period [2,3]. Assessing various preoperative risk factors and their relationship with surgical outcomes has become a crucial component of clinical decision-making. In cardiac surgery, researchers have leveraged large databases to develop numerous risk scores [3,4]. These models aim to accurately predict surgical risks, furnish information for patient counseling, aid in evaluating surgical quality, or facilitate differentiated financial reimbursement [5]. Certain patient characteristics, such as liver dysfunction, have been identified as important predictors of adverse outcomes following cardiac surgery [6]. Although the relationship between liver dysfunction and postoperative outcomes has been recognized, the exact extent of its impact on morbidity and mortality following cardiac surgery has not yet been fully elucidated [7]. However, the classical risk assessment models may not adequately capture the perioperative risks faced by these individuals. Therefore, further exploration of the interplay between liver dysfunction and surgical outcomes is warranted [8,9]. The MELD Score (Model for End-Stage Liver Disease) is calculated based on serum total bilirubin, the international normalized ratio (INR) and creatinine concentrations [10]. This scoring system was originally developed to evaluate the short-term prognoses of cirrhotic patients undergoing trans-jugular intrahepatic portosystemic shunts [11]. Subsequently, it has been used to assess mortality risk in various liver diseases regardless of the underlying etiology [12].

The MELD Score has also demonstrated excellent performance in the primary allocation of organs for liver transplantation in the United States and, after that, worldwide. Importantly, the MELD Score has been shown to have a predictive significance in determining the complications associated with certain gastrointestinal, abdominal, bariatric, musculoskeletal, cardiovascular, and urologic operations [13,14]. In these studies, a higher MELD Score was an independent risk factor for postoperative complications such as wound infections, pulmonary and cardiac issues, and urinary tract infections [15]. These surgical procedures carry a high risk of morbidity and mortality in patients with liver dysfunction [16]. The literature has reported postoperative mortality rates ranging from 5 to 85% in patients with liver dysfunction depending on the MELD Score and the type of surgery [17]. The MELD Score has also been used to evaluate the risk potential in non-surgical contexts, such as congestive heart failure and soft-tissue infections [18]. Currently, the most widely used surgical risk assessment models, such as EUROSCORE II (European System for Cardiac Operative Risk Evaluation) and STS score (Society of Thoracic Surgeons) do not incorporate any measure of liver function [18,19]. Yet, with increasing evidence demonstrating the impact of liver dysfunction on surgical outcomes, incorporating measures of liver function into risk prediction models may be necessary to enhance their precision and inform clinical decision-making [19].

The present retrospective study aimed to evaluate the effectiveness of the two scores, namely the MELD Score and EUROSCORE, in predicting postoperative mortality in patients who underwent coronary artery bypass grafting surgery as well as testing a new hypothesis that posits that the combination of the two scores creates a new score that is more effective at predicting mortality. The present research is addressed to cardiovascular surgeons, cardiologists, gastroenterologists and other medical specialties who preoperatively evaluate patients with any degree of liver dysfunction in order to issue appropriate surgical indications.

## 2. Materials and Methods

Over a period of 10 years, between 1 January 2011 and 31 December 2020, 699 patients underwent CABG surgery through the classical technique, using extracorporeal circulation, at the Institute of Cardiovascular Diseases, “Prof. Dr. George I.M. Georgescu” Iasi, Romania. The study gathered clinical characteristics and laboratory findings such as creatinine, bilirubin and INR values, ALT (Alanine aminotransferase) and AST (Aspartate aminotransferase) levels, the smoking and alcohol consumption history of the patients, as well as data needed to calculate EUROSCORE II, such as age, gender, chronic lung disease, extracardiac arteriopathy, poor mobility, previous cardiac surgery, critical preoperative state, renal impairment, diabetes on insulin, CCS angina class 4 (Canadian Cardiovascular Society angina class 4: angina at rest), LV function (left ventricular), recent MI (myocardial infarction within 90 days), pulmonary hypertension, NYHA Class (New York Heart Association Classification), surgery on thoracic aorta information, urgency of operation and weight of operation. In terms of surgery, the patients were divided into 2 groups: patients with bypass only and patients with bypass and other concomitant surgeries, such as valve prosthesis. All patients underwent echocardiographic evaluation before their surgical procedures to assess the left ventricular end-diastolic and end-systolic volumes, as well as the ejection fractions. In the study were included the patients who had all the necessary data for calculating, at the time of admission, the MELD Score, respectively, serum creatinine, bilirubin and INR values, as well as ALT and AST values, all these patients being without anticoagulant therapy (such as vitamin K antagonists, direct-acting oral anticoagulants and low-molecular-weight heparin) and presented complete medical records, including smoking and alcohol consumption history. The exclusion criteria were: incomplete data for EUROSCORE and MELD Score calculation, patients on anticoagulant therapy, as well as incomplete medical records. The application of these criteria resulted in 681 patients being included in the study, of which 589 survived and 92 died during hospitalization and in the immediate postoperative period. In order to be able to test the predictive value of mortality for the two scores (EUROSCORE and the MELD Score), a sample of 93 patients was randomly chosen from the surviving patients, using age, emergency and weight of cardiac procedures criteria to match the 92 deceased patients during hospitalization. The official formula was used to calculate the Meld Score (11.2 × (ln INR) + 0.378 × (ln TB) + 0.957 × (ln creatinine) + 0.643 [20]) and the official EUROSCORE website has been used to calculate the EUROSCORE [21]. The study was reviewed and approved by the Research Ethics Committee at Grigore T. Popa University of Medicine and Pharmacy, located in Iasi (No.163, 21 March 2022).

All the data from the study were analyzed using IBM SPSS Statistics 25 and illustrated using Microsoft Office Excel/Word 2021. Qualitative variables were written as counts or percentages and were tested between groups using a Fisher’s Exact Test. Z-tests with Bonferroni correction were used to further detail the results obtained in the contingency tables. Quantitative variables were written as means with standard deviations or medians with interquartile ranges. The normality of the quantitative variables was assessed using the Shapiro–Wilk Test. Given the non-parametric distribution of the parameters, differences in quantitative variables between survival groups were tested using the Mann–Whitney U Test.

The predictive performance of the MELD Score and EUROSCORE over mortality was measured in ROC curves by evaluation of AUC values with 95% confidence intervals and significance values. Cut-off values were calculated by considering the highest value of the Youden index; for each cut-off value, a sensitivity and a specificity was measured. Kaplan–Meier analyses were used to evaluate the differences of overall survival between patients with a high mortality risk while considering the MELD Score/EUROSCORE groups (grouped by the observed cut-offs). Overall survival was estimated as means with 95% confidence intervals in most scenarios. and differences between groups were tested using the Log-Rank Test.

Furthermore, Cox-proportional hazard models were used as both univariable and multivariable models in the prediction of mortality hazards while considering the MELD Score/EUROSCORE risks or absolute scores. The effects of each variable were calculated as hazard ratios with 95% confidence intervals along significance values. Models were tested for goodness-of-fit and significance values. DCA analyses were computed in R software (The R Project for Statistical Computing, Vienna, Austria) version 4.4.0 by using standard packages along with the dcurves, gtsummary, dplyr and tidyr packages. The highest benefit of each predictive scenario was measured by considering the current mortality prevalence (approximately 50%). The validation of the Cox-proportional hazard multivariable model using the absolute values of the MELD Score and EUROSCORE was performed using the survival and rms packages. Boot-strapping and 10-fold cross-validation methods were used to validate the initial model.

Index-corrected values were presented in comparison to the initial model to illustrate the variability of the discrimination indexes from the original model. The threshold considered for the significance level for all tests was considered to be α = 0.05.

## 3. Results

Table 1 shows the characteristics of the 185 patients, and, since most of the factors are included in the calculation of the MELD Score and EUROSCORE, the analysis will continue with the analysis of these two factors over mortality. The mean MELD Score was 6.113 ± 4.8, median = 4.588 (IQR = 2.827–8.501), being significantly higher in deceased patients (median = 8.28, IQR = 5.61–11.17) vs. survivors (median = 2.92, IQR = 2.12–4.23, *p* < 0.001), and the mean EUROSCORE was 17.65 ± 15.22, median = 12.69 (IQR = 7.42–21.52), being significantly higher in deceased patients (median = 20.7, IQR= 13.97–31.28) vs. survivors (median = 7.46, IQR = 5.30–10.46, *p* < 0.001).

Data from Figure 1 show the ROC curve analysis for the MELD Score and EUROSCORE prediction of mortality. The results show that both scores had a very good and significant performance in terms of mortality prediction: MELD Score (AUC = 0.896, 95% C.I.: 0.852–0.940, *p* < 0.001), with the cut-off being 5.54 (patients with a MELD Score ≥ 5.54 had a 76.1% sensitivity and 90.3% specificity in the mortality prediction); EUROSCORE (AUC = 0.899, 95% C.I.: 0.854–0.945, *p* < 0.001), with the cut-off being 10.37 (patients with a EUROSCORE ≥ 10.37 had a 95.7% sensitivity and 75.3% specificity in mortality prediction).

Data from Table 2 and Figure 2 and Figure 3 show the Kaplan–Meier analyses for overall survival comparison between the MELD Score/EUROSCORE groups. The results show that both factors significantly reduced the overall survival: patients with a MELD Score higher or equal to 5.54 had a lower overall survival (mean = 24.73, 95% C.I.: 18.62–30.83 vs. mean = 76.70, 95% C.I.: 12.71–140.69, *p* < 0.001), while patients with a EUROSCORE higher or equal to 10.37 had a lower overall survival (mean = 29.16, 95% C.I.: 20.53–37.8 vs. mean = 91.73, 95% C.I.: 40.17–143.3, *p* < 0.001).

Data from Table 3 show the Cox-proportional hazard models used in mortality prediction using MELD Score/EUROSCORE risks. In both the univariable and multivariable models, each score risk (MELD Score ≥ 5.54 and EUROSCORE ≥ 10.37) had a significant (*p* < 0.05) and independent effect over mortality; as such, in the multivariable model patients with a MELD Score equal or higher than 5.54 had a risk of death increased by 2.385 times (95% C.I.: 1.435–3.964) (*p* = 0.001), while patients with a EUROSCORE equal or higher than 10.37 had the risk of death increased by 8.665 times (95% C.I.: 3.091–24.293) (*p* < 0.001). The predictive nature of the scores is proven as consistent even in the adjusted multivariable model (using age and gender), where patients with a MELD Score ≥ 5.54 had the risk of death increased by 2.409 times (95% C.I.: 1.447–4.011) (*p* = 0.001) while patients with a EUROSCORE ≥ 10.37 had the risk of death increased by 8.815 times (95% C.I.: 3.143–24.718) (*p* < 0.001).

Considering the nature of the results, the hypothesis of combining the two scores for increasing the performance of mortality prediction came into existence. As such, there are two possible scenarios: having an increased risk of death either from MELD Score ≥ 5.54 or EUROSCORE ≥ 10.37 (joint scenario) or having an increased risk of death from both MELD Score ≥ 5.54 and EUROSCORE ≥ 10.37 (conditional scenario). The performance of the prediction will be evaluated in contingency tables while considering the following parameters: sensibility, specificity, positive predictive value, negative predictive value, overall accuracy and in a decision curve analysis.

Data from Table 4 and Figure 4 show the distribution of the patients according to mortality and MELD Score/EUROSCORE risks along with the joint/conditional scenarios of both scores along with the decision curve analysis. The results show the following:

-When computing performance parameters in all four scenarios, the conditional scenario (patients having both MELD Score ≥ 5.54 and EUROSCORE ≥ 10.37) had a similar accuracy in mortality prediction with the joint scenario, having an overall accuracy of 84.32% (slightly lower than EUROSCORE—85.41%); the conditional scenario gives an advantage in specificity and positive predictive value, while the joint scenario gives an advantage in sensitivity and negative predictive value.-The decision curve analysis shows that, while considering overall mortality as, as in this database, approximately 50%, the highest benefit in mortality prediction is given by the usage of EUROSCORE alone (green line), while the conditional scenario (purple line) and joint scenario (blue line) have an equal benefit, slightly lower than EUROSCORE.-However, the analysis shows that, as mortality prevalence increases, the benefit of conditional scenario usage in mortality prediction remains the highest, below the other criteria.

Data from Table 5 and Figure 5 show the Kaplan–Meier analysis for an overall survival comparison between the groups according to mortality risk based on the combined MELD Score and EUROSCORE (conditional scenario). The results show that patients with both scores at high values (MELD Score ≥ 5.54 and EUROSCORE ≥ 10.37) had a lower overall survival (mean = 21.89, 95% C.I.: 16.78–27 vs. mean = 74.76, 95% C.I.: 30.11–119.4, *p* < 0.001) in comparison to patients without this criterion.

Data from Table 6 show the Cox-proportional hazard model used in mortality prediction using MELD Score/EUROSCORE (conditional scenario). As seen in the model, patients with both risk factors present (MELD Score ≥ 5.54 and EUROSCORE ≥ 10.37) have a 4.451 times increased risk of death (95% C.I.: 2.747–7.211) (*p* < 0.001). The predictive nature of the scores has been proven as being consistent even in the adjusted multivariable model (using age and gender), where patients with both risk factors present (MELD ≥ 5.54 and EUROSCORE ≥ 10.37) have a 4.541 times increased risk of death (95% C.I.: 2.803–7.359) (*p* < 0.001). Considering the results observed in Table 3, usage of both criteria as risk factors for mortality seems to provide a more accurate prediction of mortality (with the estimate of hazard ratio being between the estimates given by the MELD Score and EUROSCORE), and the range of confidence intervals for the hazard ratio is adequate (significantly smaller than the one given by the EUROSCORE prediction).

Data from Table 7 show the Cox-proportional hazard models used in mortality prediction using the MELD/EUROSCORE absolute scores. As seen in Table 3, both scores had a significant impact over mortality prediction, each increase of 1 point in the MELD Score increases the risk of death by 1.044 times (95% C.I.: 1.008–1.081, *p* = 0.015), while an increase of 1 point in the EUROSCORE increases the risk of death by 1.030 times (95% C.I.: 1.019–1.041, *p* < 0.001). The predictive nature of the scores is proven as consistent even in the adjusted multivariable model (using age and gender), where each increase of 1 point in the MELD score increases the risk of death by 1.048 times (95% C.I.: 1.008–1.089, *p* = 0.017), while an increase of 1 point in the EUROSCORE increases the risk of death by 1.031 times (95% C.I.: 1.020–1.043, *p* < 0.001).

The bootstrapping and cross-validation of the initial models maintains the initial discrimination parameters in the bootstrap and cross-validation models, as can be seen in the Appendix A of the article, having almost the same values in the bootstrap model while, in the cross-validation models, the Nagelkerke R2 index, Slope, discrimination (D) and unreliability (U) indexes are a little increased, along with an increase in the Overall Quality (Q) and g-index. The Somers’s D (Dxy) index is variable across all methods, albeit while showing the same approximate value (approximately 0.6).

## 4. Discussion

The current study extends the prior findings by evaluating the utility of combining the MELD Score and EUROSCORE to improve mortality prediction in patients with liver dysfunction undergoing open-heart surgery, in our case coronary artery bypass graft surgery. The results suggest that using both the MELD Score and EUROSCORE in a conditional model (i.e., increased risk if both scores are elevated) provides the most accurate mortality prediction compared to using either score alone, and, to the best of our knowledge, this is the first study to use such a model to predict mortality by combining the two scores.

Liver dysfunction poses a significant health threat, with severe consequences if left unaddressed. Globally, liver diseases account for around two million deaths annually, and, in countries like Mexico, they rank as the third leading cause of death in men and the seventh in women [22]. Liver disease and fibrosis often stem from chronic inflammatory processes, which can be triggered by a range of factors, including toxins, alcohol, heavy metals, drugs, and viral infections [23]. However, the global prevalence of nonalcoholic fatty liver disease is substantially higher, affecting approximately two billion people worldwide, and cardiovascular diseases are the leading cause of mortality in these individuals, prompting the European clinical practice guidelines to recommend mandatory cardiovascular disease screenings for these patients [24]. Therefore, the total number of patients with concurrent liver dysfunction and coronary artery disease may be even greater, with many of them needing interventional diagnoses and treatment. The reduced hepatic reserve can have significant consequences for these patients, as they face an elevated risk of mortality and complications [25]. Other potential risk factors include hypo-albuminemia leading to increased bleeding and infection risk, impaired immune functions, coagulation disorders, and the development of acute kidney failure after surgery [26]. In the present case, the majority of our patients had acute or subacute liver injury associated with heart failure and hemodynamic disturbances like coagulopathy that was accompanied by renal failure. Hyperbilirubinemia and high levels of AST were also detected after admission in the group of deceased patients and patients who survived having much lower INR, creatinine, bilirubin and AST admission values, although ALT values were not significantly different in the two groups of patients, surviving and deceased. Patients who were smokers as well as those who revealed chronic alcohol consumption were more numerous in the group of patients who died compared to those who survived, reiterating both the cardiovascular and hepatic risk factors of tobacco and alcohol consumption.

The impact and effects of liver dysfunction on cardiac surgery outcomes continue to be a topic of debate based on our examination of prior research [27]. Research by Kiris et al. [14] found that patients with acute coronary syndromes who underwent PCI (percutaneous coronary intervention) and had elevated MELD scores experienced higher long-term mortality [14,28]. Additionally, Hsieh WC et al. [15] reported an association between the MELD Score and poor survival outcomes in patients with liver disease who underwent open-heart cardiac surgery [15,28]. Collectively, these studies suggest that the MELD Score may be a useful prognostic indicator for patients with underlying liver disease who undergo major cardiac procedures such as percutaneous coronary intervention or open-heart surgery [13,14,28], this being the case here, where the median of the MELD Score was considerably higher in the deceased group compared to the survivors group. Garatti et al. reported the overall 30-day mortality rate to be between 13% and 20%, depending on the extent of preoperative liver dysfunction, in patients who received cardiac surgical procedures [28]. In the present study, the mean overall survival was 39.64 days (95% C.I.: 28.26–51.02), while the median overall survival was 28 days (IQR.: 15–49 days), meaning that half of the individuals in the study survived 28 days or less and the other half survived for longer than 28 days. This is consistent with the previously documented perioperative mortality rates in liver diseases patients, ranging from 25% when the hepatic dysfunction is mild to 70% among those with the most severe levels of liver dysfunction [29]. For patients with liver dysfunction undergoing cardiac surgery with cardiopulmonary bypass, the occurrence of postoperative hepatic injury is strongly correlated with MELD Score and cardiopulmonary bypass (CPB) duration. To prevent this complication, the CPB duration should be reduced for those with MELD scores between 5 and 20, and CPB should be avoided for patients with MELD scores greater than 20 [30]. In addition to all this, the current study detected a close link between mortality and preoperative chronic lung disease, insulin-dependent diabetes, CCS angina class 4, extra-cardiac arteriopathy, moderate/poor left ventricular function, pulmonary hypertension, NYHA Class IV and, of course, the degree of difficulty of the surgery, also finding that patients who have had ≥ 3 surgical procedures performed on an emergency basis are more likely to die in the immediate postoperative period. Most studies so far have focused primarily on mortality rates in patients with acute or chronic liver cirrhosis, while few studies have examined the impact of mild-to-moderate liver dysfunction on the course and outcomes of cardiac surgical interventions, with this being the reason why we think it is very important to further explore this topic.

Kirov et al. realized a comprehensive meta-analysis, the first to include studies utilizing both the Child–Turcotte–Pugh and Model for End-Stage Liver Disease scoring systems, that revealed a strong connection between liver dysfunction and adverse outcomes following cardiac surgery. The results provide novel quantification of these relationships, highlighting the association of liver dysfunction with not only mortality but also neuro-logical events, prolonged ventilation, sepsis, bleeding, need for transfusion, and acute kidney injury [31]. Given the fact that, also in the present case, an important association be- tween MELD Score and mortality in patients undergoing CABG surgery was detected, we decided to test the predictive values of the MELD Score and EUROSCORE in regards to mortality, and the results demonstrated that both the MELD Score and EUROSCORE exhibited excellent and statistically significant performance in terms of predicting mortality.

The EUROSCORE system and its updated versions have been extensively validated and shown to reliably predict both short-term and long-term outcomes following cardiac surgery. Compared to other risk assessment tools, EUROSCORE and EUROSCORE II are more straightforward to calculate and are widely adopted in clinical practice [32]. However, some research has suggested that incorporating additional factors, such as measures of functional status, could further enhance the predictive capabilities of these models [33]. The limitations of this risk system are found in encompassing all relevant comorbidities due to inherent statistical constraints. Conditions like hepatic dysfunction, which were uncommon in the initial dataset, did not reach the threshold of statistical significance [34]. However, emerging evidence suggests that the Child–Turcotte–Pugh and MELD scores may hold greater predictive value for both early and late mortality [35], this retrospective analysis being one of them.

This present study evaluated the predictive performance of the well-established MELD Score and EUROSCORE II systems in estimating mortality risk among patients hospitalized for heart failure. According to the Cox-proportional hazard model shown, patients with both risk factors present (MELD Score ≥ 5.54 and EUROSCORE ≥ 10.37) have a 4.45 times increased risk of death (95% C.I.: 2.74–7.21, *p* < 0.001), and the hazard ratio’s confidence interval range is satisfactory. Our findings demonstrated that both scoring models were significantly associated with mortality in both univariable and multivariable analyses. Importantly, by integrating the two scoring systems in a conditional scenario, we were able to achieve a high accuracy in mortality prediction, with an overall accuracy of 84.32%.

The limitations of this study include its retrospective, single-center design and the relatively small sample size. Additionally, the specific underlying etiologies of liver disease were not available, which may have influenced patient outcomes. These patients may have unaccounted-for additional risk factors that are not fully captured by the EUROSCORE II results.

## 5. Conclusions

The added benefit of combining the two scoring systems in a conditional scenario may be the fact that they capture different aspects of a patient’s clinical status. The MELD Score primarily reflects liver function, while the EUROSCORE incorporates factors related to cardiac surgery risk. By considering both of these factors, the conditional scenario may provide a more comprehensive assessment of a patient’s overall clinical condition and risk of mortality. Therefore, we consider that the team working on the new EUROSCORE should consider the option of analyzing these results with a view to including them in their study.

## Figures and Tables

**Figure 1 biomedicines-13-00154-f001:**
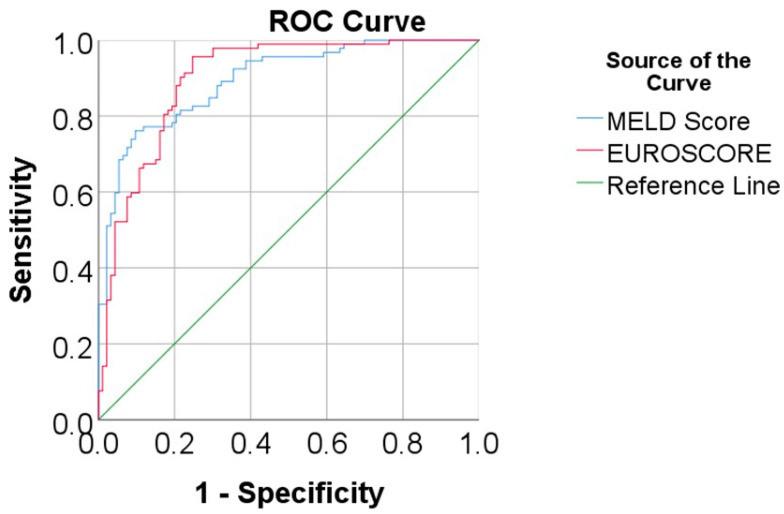
ROC curve analysis for the MELD Score and EUROSCORE prediction of mortality.

**Figure 2 biomedicines-13-00154-f002:**
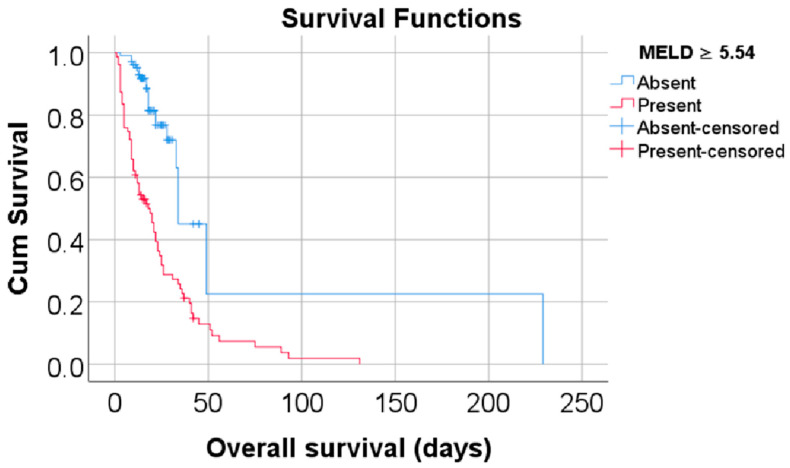
Kaplan–Meier curve for an overall survival comparison between the MELD Score groups.

**Figure 3 biomedicines-13-00154-f003:**
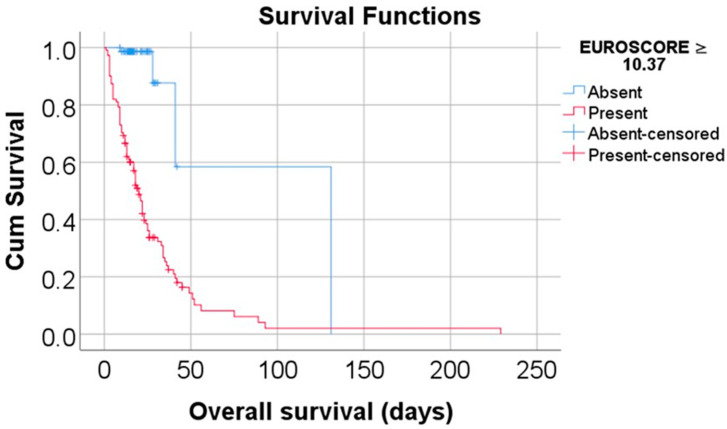
Kaplan–Meier curve for an overall survival comparison between the EUROSCORE groups.

**Figure 4 biomedicines-13-00154-f004:**
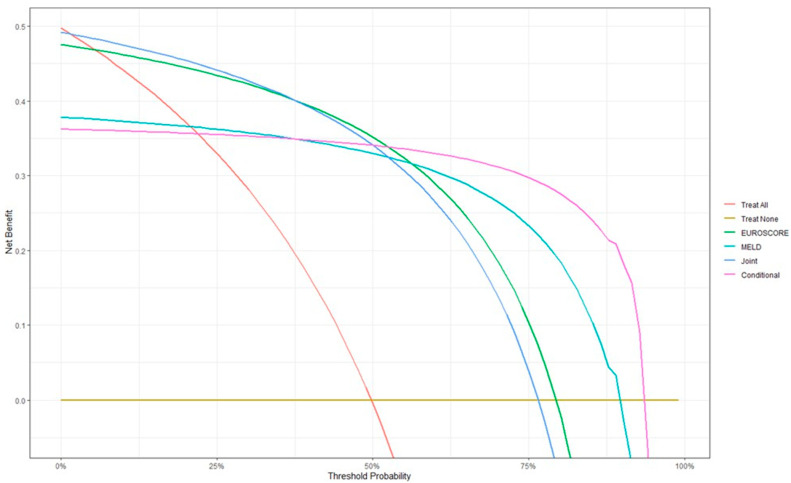
Decision curve analysis in mortality prediction.

**Figure 5 biomedicines-13-00154-f005:**
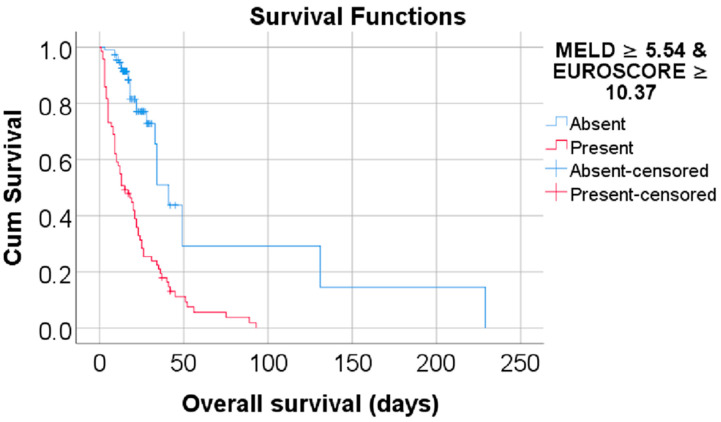
Kaplan–Meier curve for an overall survival comparison between groups according to mortality risk based on a combined MELD Score and EUROSCORE (conditional scenario).

**Table 1 biomedicines-13-00154-t001:** Characteristics of the analyzed patients.

Variable	Survivors (*n* = 93)	Deceased (*n* = 92)	*p*
Age (years) (median (IQR))	70 (63–74.5)	68 (63–74)	0.828 *
Gender (male) (Nr., %)	65 (69.9%)	68 (73.9%)	0.624 **
Smoking (Nr., %)	15 (16.1%)	33 (35.9%)	0.003 **
Alcohol consumption (Nr., %)	28 (30.1%)	42 (45.7%)	0.034 **
Chronic lung disease (Nr., %)	10 (10.8%)	23 (25%)	0.013 **
Extracardiac arteriopathy (Nr., %)	22 (23.7%)	81 (88%)	<0.001 **
Poor mobility (Nr., %)	10 (10.8%)	24 (26.1%)	0.008 **
Previous cardiac surgery (Nr., %)	0 (0%)	1 (1.1%)	0.497 **
Critical preoperative state (Nr., %)	0 (0%)	14 (15.2%)	<0.001 **
Renal impairment (Nr., %)			
Absent	60 (64.5%)	47 (51.1%)	0.182 **
Moderate	24 (25.8%)	31 (33.7%)	
Severe	7 (7.5%)	13 (14.1%)	
Dialysis necessity	2 (2.2%)	1 (1.1%)	
Renal impairment (Nr., %)	33 (35.5%)	45 (48.9%)	0.075 **
Diabetes on Insulin (Nr., %)	20 (21.5%)	42 (45.7%)	0.001 **
CCS angina class 4 (Nr., %)	67 (72%)	90 (97.8%)	<0.001 **
LV function (Nr., %)			
Good	67 (72%)	25 (27.2%)	<0.001 **
Moderate	24 (25.8%)	55 (59.8%)	
Poor	2 (2.2%)	12 (13%)	
LV function recoded (Nr., %)			
Good	67 (72%)	25 (27.2%)	<0.001 **
Moderate/poor	26 (28%)	67 (72.8%)	
Recent MI (Nr., %)	62 (66.7%)	81 (88%)	0.001 **
Pulmonary hypertension (Nr., %)			
Absent	89 (95.7%)	77 (83.7%)	0.009 **
Moderate PH	4 (4.3%)	14 (15.2%)	
Severe PH	0 (0%)	1 (1.1%)	
Pulmonary hypertension (Nr., %)	4 (4.3%)	15 (16.3%)	0.008 **
NYHA Class (Nr., %)			
Class III	4 (4.3%)	14 (15.2%)	<0.001 **
Class IV	56 (60.2%)	92 (100%)	
Surgery on thoracic aorta (Nr., %)	1 (1.1%)	0 (0%)	1.000**
Urgency of operation (Nr., %)			
Elective	9 (9.7%)	9 (9.8%)	1.000 **
Urgent	35 (37.6%)	35 (38%)	1.000 **
Emergency	49 (52.7%)	48 (52.2%)	1.000 **
Non-elective operation (Nr., %)	84 (90.3%)	83 (90.2%)	1.000 **
Weight of operation (Nr., %)			
Two Procedures	7 (7.5%)	6 (6.5%)	1.000 **
Three Procedures	86 (92.5%)	86 (93.5%)	
Surgery type (Nr., %)			
Only bypass	60 (64.5%)	69 (75%)	0.150 **
Bypass + other interventions	33 (35.5%)	23 (25%)	
AST (U/L) (Median (IQR))	25 (20–38.5)	31 (21–51.25)	0.022 *
ALT (U/L) (Median (IQR))	30 (21–54.5)	28 (19.25–42.75)	0.414 *
Creatinine (mg/dL) (Median (IQR))	0.87 (0.79–1.02)	1.11 (0.96–1.28)	<0.001 *
Bilirubin (mg/dL) (Median (IQR))	0.53 (0.42–0.74)	0.81 (0.67–1.02)	<0.001 *
INR (Median (IQR))	0.98 (0.93–1.07)	1.10 (1.02–1.25)	<0.001 *
MELD Score (Median (IQR))	2.92 (2.12–4.23)	8.28 (5.61–11.17)	<0.001 *
EUROSCORE (Median (IQR))	7.46 (5.30–10.46)	20.7 (13.97–31.28)	<0.001 *
Overall survival (days) Mean (95% C.I.), (Median (IQR))	39.64 (28.26–51.02), 28 (15–49)		-

INR—international normalized ratio, CCS—Canadian Cardiovascular Society, LV—left ventricular, MI—myocardial infarction, NYHA—New York Heart Association, MELD—Model for End-Stage Liver Disease, EUROSCORE—European System for Cardiac Operative Risk Evaluation. ***** Statistical comparisons were made using the Mann–Whitney U test at a significance level of *p* < 0.05, and the results were reported as medians with interquartile ranges. ****** A Fisher’s Exact Test was used for the analysis of categorical variables at a *p*-value < 0.05 significance level.

**Table 2 biomedicines-13-00154-t002:** Kaplan–Meier analyses for an overall survival comparison between the MELD Score/EUROSCORE groups.

MELD Score ≥ 5.54	Mean (95% C.I.)	*p* *
Absent	76.70 (12.71–140.69)	<0.001
Present	24.73 (18.62–30.83)	
**EUROSCORE ≥ 10.37**	**Mean (95% C.I.)**	***p* ***
Absent	91.73 (40.17–143.3)	<0.001
Present	29.16 (20.53–37.8)	

* Log-Rank Test.

**Table 3 biomedicines-13-00154-t003:** Cox-proportional hazard models used in mortality prediction using MELD/EUROSCORE risks.

Parameter	Univariable		Multivariable	
	HR (95% C.I.)	*p*	HR (95% C.I.)	*p*
MELD ≥ 5.54	4.062 (2.467–6.687)	<0.001	2.385 (1.435–3.964)	0.001
EUROSCORE ≥ 10.37	12.56 (4.591–34.359)	<0.001	8.665 (3.091–24.293)	<0.001
**Adjusted Cox-Proportional Multivariable Hazard Model with Age and Gender ***				
**Parameter**	**HR (95% C.I.)**		** *p* **	
MELD ≥ 5.54	2.409 (1.447–4.011)		0.001	
EUROSCORE ≥ 10.37	8.815 (3.143–24.718)		<0.001	
Age	0.981 (0.962–1.001)		0.057	
Gender (Male)	0.931 (0.573–1.513)		0.773	

* Adjusted for age and gender.

**Table 4 biomedicines-13-00154-t004:** The distribution of the patients according to the mortality and MELD Score/EUROSCORE risks along with joint/conditional scenarios of both scores.

MELD Score ≥ 5.54/Death	Survivors		Deceased		*p* *
	Nr.	%	Nr.	%	
Absent	84	90.3%	22	23.9%	<0.001
Present	9	9.7%	70	76.1%	
Se = 76.09%, Sp = 90.32%, PPV = 88.61%, NPV = 79.25%, Accuracy = 83.24%					
**EUROSCORE ≥ 10.37/Death**	**Survivors**		**Deceased**		***p* ***
	**Nr.**	**%**	**Nr.**	**%**	
Absent	70	75.3%	4	4.3%	<0.001
Present	23	24.7%	88	95.7%	
Se = 95.65%, Sp = 75.27%, PPV = 79.28%, NPV = 94.59%, Accuracy = 85.41%					
**MELD Score ≥ 5.54 OR EUROSCORE ≥ 10.37/Death (Joint Scenario)**	**Survivors**		**Deceased**		***p* ***
	**Nr.**	**%**	**Nr.**	**%**	
Absent	65	69.9%	1	1.1%	<0.001
Present	28	30.1%	91	98.9%	
Se = 98.91%, Sp = 69.89%, PPV = 76.47%, NPV = 98.48%, Accuracy = 84.32%					
**MELD Score ≥ 5.54 AND EUROSCORE ≥ 10.37/Death (Conditional Scenario)**	**Survivors**		**Deceased**		***p* ***
	**Nr.**	**%**	**Nr.**	**%**	
Absent	89	95.7%	25	27.2%	<0.001
Present	4	4.3%	67	72.8%	
Se = 72.83%, Sp = 95.70%, PPV = 94.37%, NPV = 78.07%, Accuracy = 84.32%					

* Fisher’s Exact Test.

**Table 5 biomedicines-13-00154-t005:** Kaplan–Meier analysis for overall survival comparison between groups according to mortality risk based on the combined MELD score and EUROSCORE (conditional scenario).

Mortality Risk (Conditional Scenario) (MELD ≥ 5.54 and EUROSCORE ≥ 10.37)	Mean (95% C.I.)	*p* *
Absent	74.76 (30.11–119.4)	<0.001
Present	21.89 (16.78–27)	

* Log-Rank Test.

**Table 6 biomedicines-13-00154-t006:** The Cox-proportional hazard model used in mortality prediction using MELD/EUROSCORE (conditional scenario).

Parameter	HR (95% C.I.)	*p*
MELD ≥ 5.54 and EUROSCORE ≥ 10.37	4.451 (2.747–7.211)	<0.001
**Adjusted Cox-Proportional Multivariable Hazard Model with Age and Gender ***		
**Parameter**	**HR (95% C.I.)**	*p*
MELD ≥ 5.54 and EUROSCORE ≥ 10.37	4.541 (2.803–7.359)	<0.001
Age	0.982 (0.962–1.001)	0.069
Gender (Male)	0.917 (0.565–1.487)	0.725

* Adjusted for Age and Gender.

**Table 7 biomedicines-13-00154-t007:** Cox-proportional hazard models used in mortality prediction using MELD/EUROSCORE absolute scores.

Parameter	Univariable		Multivariable	
	HR (95% C.I.)	*p*	HR (95% C.I.)	*p*
MELD Score	1.087 (1.053–1.121)	<0.001	1.044 (1.008–1.081)	0.015
EUROSCORE	1.035 (1.026–1.045)	<0.001	1.030 (1.019–1.041)	<0.001
**Adjusted Cox-proportional multivariable hazard model with age and gender ***				
**Parameter**	**HR (95% C.I.)**		** *p* **	
MELD Score	1.048 (1.008–1.089)		0.017	
EUROSCORE	1.031 (1.020–1.043)		<0.001	
Age	1.009 (0.987–1.031)		0.442	
Gender (Male)	1.105 (0.672–1.819)		0.694	

* Adjusted for age and gender.

## Data Availability

The data were obtained from the Institute of Cardiovascular Diseases and “Prof. Dr. George I.M. Georgescu” of Iasi, and they were extracted from electronic medical records and printed operating procedures following approval from its Institutional Review Board.

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
