# Peer review of "European System for Cardiac Operative Risk Evaluation II and Liver Dysfunction"

_biomedicines, 2025, doi:10.3390/biomedicines13010154_

Round 1

Reviewer 1 Report

Comments and Suggestions for Authors

The article explores the combination of 2 risk scores, one cardiovascular and the other for liver failure. Some considerations should be included to better assess the results.

Major Comments

The multivariate models used support databases with a slight imbalance such as the one presented here: 589 survivors and 92 deaths in the postoperative period and with a 10-year follow-up. It is advisable to have a training and testing strategy for the model that allows the incorporation of cross-validation techniques.

A dichotomization of the scores a priori today is not recommended since the cut-off points are sensitive populations.

The estimated cut-off point for EUROSCORE II may be questionable.

You could build a model that combines the EUROSCORE II and MELD score variables instead of treating them as dichotomous variables in a multivariate model.

Strategies in addition to the randomized selection of survivors could improve performance in the metrics analyzed. These should at least be provided as supplementary data if they do not outperform the random selection performance

Minor comments

If maintained after all the previous requests, a survival curve with the conditional scenario could be recommended 

Author Response

The multivariate models used support databases with a slight imbalance such as the one presented here: 589 survivors and 92 deaths in the postoperative period and with a 10-year follow-up. It is advisable to have a training and testing strategy for the model that allows the incorporation of cross-validation techniques.

A dichotomization of the scores a priori today is not recommended since the cut-off points are sensitive populations.

The estimated cut-off point for EUROSCORE II may be questionable.

You could build a model that combines the EUROSCORE II and MELD score variables instead of treating them as dichotomous variables in a multivariate model.

Strategies in addition to the randomized selection of survivors could improve performance in the metrics analyzed. These should at least be provided as supplementary data if they do not outperform the random selection performance.

Response 1: Thank you very much for the valuable guidance on our research. We have taken them into account and we redid the study by randomly choosing 93 survivors  from the total of 589 surviving patients, using age, emergency and weight of cardiac procedures criteria to match the 92 deceased patients and we redid the statistical tests as can be seen on pages 4-10.

Best regards!

Reviewer 2 Report

Comments and Suggestions for Authors

There is a crucial problem with this MS, the majority of patients who died were operated as urgent and emergency setting which can seriously influence liver and renal function and these two groups are incomparable. 

The choice of control group - survived group, must be matched at least using age, emergency and weight of the cardiac procedures.   

Comments on the Quality of English Language

There are several imprecisions for some English terms and in my opinion this MS needs English editing after revision. 

Author Response

Comments 1: 

There is a crucial problem with this MS, the majority of patients who died were operated as urgent and emergency setting which can seriously influence liver and renal function and these two groups are incomparable. 

The choice of control group - survived group, must be matched at least using age, emergency and weight of the cardiac procedures.   

Response 1: Thank you very much for the valuable indications regarding our study. We have taken them into account and redone the study, using a sample of 93 patients randomly chosen from the surviving patients, using age, emergency and weight of cardiac procedures criteria to match the 92 deceased patients during hospitalization.

2. Comments on the Quality of English Language

There are several imprecisions for some English terms and in my opinion this MS needs English editing after revision. 

Response 2: Thank you for pointing this out for us, we'll access the MDPI Author Services to resolve this issue.

Best regards!

Reviewer 3 Report

Comments and Suggestions for Authors

Specific recommendations for the revision:

Specific Comments to authors:

·       In the Materials and Methods section: The authors mentioned ONLY Exclusion criteria (incomplete data for EUROSCORE and MELD Score calculation, patients on anticoagulant therapy, and incomplete medical records), but they didn’t mention any specific Inclusion criteria. The authors didn’t mention the substance use history (smoking and alcohol consumption history) of patients as (a combined effect of higher age and substance use) it can also be one of the key factors that may affect the severity of disease and mortality.

·       Though The MELD Score (Model for End-Stage Liver Disease) is calculated based on serum total bilirubin, international normalized ratio (INR) and creatinine concentrations. However, few other liver function tests, such as levels of Alanine transaminase (ALT) and Aspartate aminotransferase (AST) are also useful parameters or markers that need to be incorporated in Table 1 (Characteristics of the analyzed patients), if possible.

Author Response

Comments: 

In the Materials and Methods section: The authors mentioned ONLY Exclusion criteria (incomplete data for EUROSCORE and MELD Score calculation, patients on anticoagulant therapy, and incomplete medical records), but they didn’t mention any specific Inclusion criteria. The authors didn’t mention the substance use history (smoking and alcohol consumption history) of patients as (a combined effect of higher age and substance use) it can also be one of the key factors that may affect the severity of disease and mortality.

  • Though The MELD Score (Model for End-Stage Liver Disease) is calculated based on serum total bilirubin, international normalized ratio (INR) and creatinine concentrations. However, few other liver function tests, such as levels of Alanine transaminase (ALT) and Aspartate aminotransferase (AST) are also useful parameters or markers that need to be incorporated in Table 1 (Characteristics of the analyzed patients), if possible.

Response: Thank you very much for the valuable guidance on our study. We have taken them into consideration and have re-done the analysis to include ASL, ALT values as well as substance use history (smoking and alcohol consumption history) in both the deceased and surviving groups. We also revised the inclusion criteria for patients in the study to be clearer, as can be seen at page 3.

Best regards!

Round 2

Reviewer 1 Report

Comments and Suggestions for Authors

The authors did a good job adapting the requested survival analyses. As expected, the cut-off points changed with the adjustment but remained in a similar range.

As minor comments, I reiterate the suggestion to incorporate the survival curves for the combined assessment and the conditional version of both scores.

Author Response

Comment 1:The authors did a good job adapting the requested survival analyses. As expected, the cut-off points changed with the adjustment but remained in a similar range.

Response 1: Thank you for your appreciation.

Comment 2: As minor comments, I reiterate the suggestion to incorporate the survival curves for the combined assessment and the conditional version of both scores.

Response 2: Than you for the suggestion. The analysis was added as Table V and Figure 5, pages 9 and10, along with the interpretation.

Best Regards!

Reviewer 2 Report

Comments and Suggestions for Authors

This is revised manuscript.

I would like to thank you authors because they accepted the main suggestion from my review to do matching for deceased patients and controls and to make major revision.

I have several questions and suggestions:

1.    In the abstract uniform the decimal numbers, use 2, not 3.

2.    Raw 35 in abstract, authors have written “sensibility”, instead of sensitivity? Please clarify.

3.    For the cut off points for MELD and EUROSCORE II it should be more effective to use 1 decimal number: 5.5 and 10.4. for instance.

4.    In the methodology section, the authors explain what consists of MELD score, and mention transaminases, however, transaminases are not part of the MELD score. Since there were some updates in the MELD score, authors must precisely defined what score did they use, and to reference that.

5.    When the components of MELD score were measured? Just before surgery? At admission to hospital? It is important to use measurements which were taken in the same time, very close to surgery, one or a few days apart.

6.    In the results section, it is not necessary to re-write all the data from the table 1. Authors can leave in the text only the most important numbers, such as median values of MELD and EUROSCORE II.

7.    The order of variables in rows are inappropriate, please use the following order: age, gender, comorbities, kind of surgery, elements of MELD score, and scores at the end of the table 1.

8.    Please erase the table 2 (it is not necessary), and write the AUCs with 95%CI in the text.

9.    I don’t understand Kaplan-Meyers curves?! According to them all patients died.

10. For the multivariable analysis please write in the text where you present results and in the footnote of the table 4 what variables were used for the adjustment.

11. Please describe in more details the figure V, what the lines presented and who to interpret this figure.

12.  Put table 6 only in the text.

13. Table 7 and 8 put in the supplementary material, these statistics and data is not necessary for the understanding the main results.

14.  In the disscusion section, authors has focused on cirrhosis, however, how many patients in their cohorts actualy had cirrhosis? There is no data about the history of cirrhosis. The majority of their patients probably had acute or subacute liver injury associated with heart failure and hemodynamic disturbances.

Comments on the Quality of English Language

I think English could be better. 

Author Response

I would like to thank you authors because they accepted the main suggestion from my review to do matching for deceased patients and controls and to make major revision.

I have several questions and suggestions:

  1. In the abstract uniform the decimal numbers, use 2, not 3.
  2. Raw 35 in abstract, authors have written “sensibility”, instead of sensitivity? Please clarify.
  3. For the cut off points for MELD and EUROSCORE II it should be more effective to use 1 decimal number: 5.5 and 10.4. for instance.
  4. In the methodology section, the authors explain what consists of MELD score, and mention transaminases, however, transaminases are not part of the MELD score. Since there were some updates in the MELD score, authors must precisely defined what score did they use, and to reference that.
  5. When the components of MELD score were measured? Just before surgery? At admission to hospital? It is important to use measurements which were taken in the same time, very close to surgery, one or a few days apart.
  6. In the results section, it is not necessary to re-write all the data from the table 1. Authors can leave in the text only the most important numbers, such as median values of MELD and EUROSCORE II.
  7. The order of variables in rows are inappropriate, please use the following order: age, gender, comorbities, kind of surgery, elements of MELD score, and scores at the end of the table 1.
  8. Please erase the table 2 (it is not necessary), and write the AUCs with 95%CI in the text.
  9. I don’t understand Kaplan-Meyers curves?! According to them all patients died.
  10. For the multivariable analysis please write in the text where you present results and in the footnote of the table 4 what variables were used for the adjustment.
  11. Please describe in more details the figure V, what the lines presented and who tointerpret this figure.
  12. Put table 6 only in the text.
  13. Table 7 and 8 put in the supplementary material, these statistics and data is not necessary for the understanding the main results.
  14. In the disscusion section, authors has focused on cirrhosis, however, how many patients in their cohorts actualy had cirrhosis? There is no data about the history of cirrhosis. The majority of their patients probably had acute or subacute liver injury associated with heart failure and hemodynamic disturbances.

Comments on the Quality of English Language

I think English could be better

Response: Thank you very much for your comments and suggestions. Please see the attachment with our responses.

Best regards!

Round 3

Reviewer 2 Report

Comments and Suggestions for Authors

Authors have answer all my questions and accept the most important suggestions.